# Two Different Isocitrate Dehydrogenases from *Pseudomonas aeruginosa*: Enzymology and Coenzyme-Evolutionary Implications

**DOI:** 10.3390/ijms241914985

**Published:** 2023-10-07

**Authors:** Xuefei Chen, Wei Wei, Wei Xiong, Shen Wu, Quanchao Wu, Peng Wang, Guoping Zhu

**Affiliations:** Anhui Provincial Key Laboratory of Molecular Enzymology and Mechanism of Major Diseases and Key Laboratory of Biomedicine in Gene Diseases and Health of Anhui Higher Education Institutes, Anhui Normal University, Wuhu 241000, China; cxf231222@ahnu.edu.cn (X.C.); weiwei178185@ahnu.edu.cn (W.W.); xiongwei@anhu.edu.cn (W.X.); wshen18@ahnu.edu.cn (S.W.); philwu@ahnu.edu.cn (Q.W.)

**Keywords:** isocitrate dehydrogenase, *Pseudomonas aeruginosa*, coenzyme specificity, phosphorylation, evolution

## Abstract

*Pseudomonas aeruginosa* PAO1, as an experimental model for Gram-negative bacteria, harbors two NADP^+^-dependent isocitrate dehydrogenases (NADP-IDHs) that were evolved from its ancient counterpart NAD-IDHs. For a better understanding of PaIDH1 and PaIDH2, we cloned the genes, overexpressed them in *Escherichia coli* and purified them to homogeneity. PaIDH1 displayed higher affinity to NADP^+^ and isocitrate, with lower Km values when compared to PaIDH2. Moreover, PaIDH1 possessed higher temperature tolerance (50 °C) and wider pH range tolerance (7.2–8.5) and could be phosphorylated. After treatment with the bifunctional PaIDH kinase/phosphatase (PaIDH K/P), PaIDH1 lost 80% of its enzymatic activity in one hour due to the phosphorylation of Ser115. Small-molecule compounds like glyoxylic acid and oxaloacetate can effectively inhibit the activity of PaIDHs. The mutant PaIDH1-D346I347A353K393 exhibited enhanced affinity for NAD^+^ while it lost activity towards NADP^+^, and the Km value (7770.67 μM) of the mutant PaIDH2-L589 I600 for NADP^+^ was higher than that observed for NAD^+^ (5824.33 μM), indicating a shift in coenzyme specificity from NADP^+^ to NAD^+^ for both PaIDHs. The experiments demonstrated that the mutation did not alter the oligomeric state of either protein. This study provides a foundation for the elucidation of the evolution and function of two NADP-IDHs in the pathogenic bacterium *P. aeruginosa*.

## 1. Introduction

The Gram-negative bacterium *P. aeruginosa* is widely distributed across various ecological niches, including soil, water environments and humans, owing to its remarkable physiological and genetic adaptability [1,2]. It is recognized as an opportunistic pathogen capable of causing a range of diseases in susceptible populations, such as cystic fibrosis and meningitis. The inherent resistance of *P. aeruginosa* to antiseptics and antibiotics can be attributed partly to its low outer membrane permeability [3]. *P. aeruginosa* secretes a vast array of virulence factors into the environment, which are closely associated with its pathogenicity [4,5]. Many of these virulence factors can subvert host defenses, compromise host tissues or enhance bacterial survival mechanisms. Novel therapeutic strategies are urgently needed to address this pressing issue [6].

The key enzyme IDH catalyzes the oxidative decarboxylation of isocitrate in the TCA cycle, leading to the production of α-ketoglutarate. This process not only yields energy but also provides biosynthetic precursors for metabolism [7]. Based on their coenzyme dependence, IDHs can be categorized into two categories: NADP-IDH and NAD-IDH. NADP-IDH generates NADPH through interacting with the 2′-phosphate of NADP^+^, thereby supplying reducing power to the organism, while NAD-IDH produces NADH by interacting with both the 2′- and 3′-hydroxyl groups of NAD^+^, which participate in material and energy metabolism [8]. The ancient IDH phenotype is NAD-IDH, and the emergence of NADP-IDH as an adaptation to poor carbon sources occurred approximately 3.5 billion years ago [7]. NAD-IDH is present in the mitochondria of a wide range of eukaryotes, as well as some archaea and bacteria, such as *Pyrococcus furiosus* (PfIDH), *Zymomonas mobilis* (ZmIDH), yeast and certain animals and plants [9,10,11,12,13,14,15,16,17,18,19]. The NADP-IDH enzyme is ubiquitously present in both prokaryotes and eukaryotic organelles, as well as cytosol, playing a crucial role in catalyzing the formation of NADH, which is essential for energy metabolism. Moreover, it facilitates ATP production through the respiratory chain, thereby providing essential energy and carbon skeletons to living organisms [10]. NADP-IDH also serves as a catalyst for the generation of NADPH, a crucial reducing agent involved in biosynthesis. Additionally, it assumes a critical function in cellular antioxidant defense and detoxification systems against reactive oxygen species (ROS) [20,21]. The conversion of *E. coli*’s NADP-IDH to NAD-IDH through amino acid substitution at the D^344^I^345^A^351^K^391^S^393^ site has been reported [8]. EcIDH is the most extensively studied enzyme, with its catalytic regulation mechanism, crystal structure and enzymatic properties being thoroughly investigated [22,23,24,25,26]. Further investigation into the evolutionary origins of two NADP-IDHs in *P. aeruginosa* PAO1 and their impact on the pathogenic bacterium’s metabolic processes is necessary.

Recently, IDHs have been taxonomically classified into three distinct phylogenetic subfamilies: type I, type II and type III [27,28,29,30]. Type I IDHs comprise bacterial homodimer NAD(P)-IDHs, eukaryotic mitochondrial heterooligomer NAD-IDHs and homotetramer NAD-IDHs. Type II IDHs encompass bacterial and eukaryotic homodimer NADP-IDHs as well as homodimer NAD-IDHs from marine eukaryotic algae. Previous research has indicated that type III IDHs are predominantly composed of NADP-IDHs, with a few exceptions being the monomeric IDHs found in prokaryotes such as *Azotobacter vinelandii*, *Corynebacterium glutamicum* and *Streptomyces lividans* [31,32,33]. However, it has recently been found that the type III IDH subfamily, as exemplified by *Acinetobacter baumannii* and *Mycobacterium tuberculosis* IDHs, also harbors a homodimeric IDH consisting of two monomeric IDH-like subunits [30,34].

Previous studies and bioinformatics analyses have demonstrated that *P. aeruginosa* PAO1 possesses IDH1, IDH2, IDH K/P and ICL, indicating the covalent regulation of glyoxylate bypass in its metabolic activity. In this study, we cloned and overexpressed genes encoding IDH1, IDH2 and IDH K/P from *P. aeruginosa* PAO1 as fused proteins with the 6xHis-tag in *E. coli*. Subsequently, the enzymatic properties, coenzyme specificity and phosphorylation reactions of these recombinant proteins were comprehensively characterized.

## 2. Results

### 2.1. Sequence and Structure Analysis

Analysis of the *P. aeruginosa* PAO1 genome revealed the presence of two IDHs: PaIDH1 with 418 amino acids and PaIDH2 with 741 amino acids. Type I NADP-IDHs from *Pseudomonas fluorescens*, *E. coli* and *Serratia symbiotica* exhibited sequence similarity of over 75% to PaIDH1. The conserved key amino acids involved in the coenzyme binding for type I IDHs (EcIDH and BsIDH) were also present in PaIDH1 (Figure 1A), suggesting that it was also a type I NADP-IDH. Comparing PaIDH2 with AvIDH, CgIDH, MtIDH2 and AbIDH2 revealed that the sequence identity of PaIDH2 with AvIDH and MtIDH2 was as high as 65%, and the sequence similarity of PaIDH2 with CgIDH and AbIDH2 was approximately 50%. All four enzymes belonged to type III NADP-IDHs, and key amino acids involved in coenzyme binding were conserved between AvIDH, CgIDH and PaIDH2 (Figure 1B). These findings suggest that PaIDH2 may be classified as a type III NADP-IDH. Only a limited number of prokaryotes, such as *A. baumannii* and *M. tuberculosis*, exhibit the presence of both IDHs. In these organisms, AbIDH2 and MtIDH2 are classified as type III NADP-IDHs, while AbIDH1 is categorized as a type I NADP-IDH and MtIDH1 falls under type II NADP-IDH classification. These distinct classifications reflect the diverse evolutionary histories of IDHs.

The phylogenetic analysis results showed that the IDHs could be classified into three phylogenetic subfamilies (Figure 2), supporting our hypothesis that PaIDH1 belongs to type I IDHs and PaIDH2 belongs to type III IDHs. This suggested distinct evolutionary paths for PaIDH1 and PaIDH2.

By comparing the structure of EcIDH in complex with the NADP^+^ ligand (PDB code: 4AJ3) to that of PaIDH1, it was found that Lys346, Tyr347, Val353 and Tyr393 in PaIDH1 perfectly corresponded to Lys344, Tyr345, Val351 and Tyr391 in EcIDH (Figure 3A). These amino acid residues were crucial determinants for the coenzyme specificity of EcIDH. The literature reported the structural elucidation of PaIDH2 binding to the small molecule NADP^+^ (6G3U). Subsequent analysis revealed that His589, Arg600 and Arg649 were crucial amino acid residues involved in the coenzyme binding of PaIDH2 (Figure 3B). These findings aligned with the results obtained from the sequence alignment analysis, further supporting that PaDH1 belongs to type I NADP-IDHs while PaIDH2 belongs to type III NADP-IDHs.

### 2.2. Overexpression and Purification of Recombinant PaIDH1, PaIDH2 and PaIDH K/P

PET-28b (+)-PaIDH1 with 6×His-tag was heterogeneously expressed in *E. coli* Rosetta (DE3) and purified by affinity ion chromatography. The SDS-PAGE results showed that PaIDH1 was expressed in a soluble supernatant and its subunit molecular weight was 45 kDa, which was consistent with the predicted theoretical molecular weight. Gel filtration was performed to find that the elution volume of PaIDH1 was 14.28 mL, and its natural molecular weight was estimated to be about 75.4 kDa, suggesting a homodimeric structure in solution (Figure 4A), and previous studies confirmed our hypothesis [35].

PET-28b (+)-PaIDH2 was heterogeneously expressed and purified in the same way in which PaIDH2 was expressed and purified. SDS-PAGE showed that the molecular weight of a single subunit of PaIDH2 was 81 kDa, and, due to this, PaIDH2 was conjectured to be a typical monomeric IDH. However, the elution volume of PaIDH2 was 12.58 mL and the molecular mass of PaIDH2 was speculated to be 176.3 kDa by gel filtration chromatography (Figure 4B). This means that PaIDH2 is not a monomer in the traditional sense, consistent with previous findings on AbIDH2. According to previous studies, PaIDH2 is composed of two asymmetric domains, with small and overlapping interfaces, and exists in the form of a trimer or elongated dimer in solution [35].

pET-28b (+)-PaIDH K/P was heterogeneously expressed and purified by the same method. SDS-PAGE showed that the molecular weight of a single subunit was 66 kDa, and gel filtration chromatography showed that the molecular weight of PaIDH K/P was 67 kDa (Figure 4C). The data indicated that PaIDH K/P was a monomer in the same polymerized form as EcIDH K/P [22,36].

It has been reported that in *M. tuberculosis*, MtIDH2 exists as a homologous tetramer in 0.1 M NaCl buffer, while it undergoes depolymerization into dimers in 1 M NaCl buffer [34]. To further investigate the polymerization form of PaIDH2, we varied the salt concentration in the solution. SDS-PAGE analysis revealed that the molecular weight of a single protein subunit remained constant across different salt concentrations. However, at a concentration of 0.1 M NaCl, PaIDH2 eluted at a volume of 11.12 mL with an injection protein concentration of 3.2 mg/mL, indicating the presence of a tetrameric form with a molecular weight of 320 kDa. At a concentration of 1 M NaCl, the elution volume was observed at 11.57 mL with an injection protein concentration of 2.7 mg/mL, suggesting its trimeric nature and a molecular weight of 260 kDa. Furthermore, at a higher salt concentration (3 M NaCl), the elution volume shifted to 11.75 mL while maintaining an injection protein concentration of 3.0 mg/mL, confirming its trimeric state and molecular weight as approximately 258 kDa (Figure 4D). The distinct polymerization forms observed under varying salt concentrations suggested potential functional adaptations of PaIDH2 in response to changes in ionic strength; however, the underlying relationship between these diverse protein conformations remains elusive.

### 2.3. Effects of pH, Temperature and Various Compounds on the Activity of PaIDH1 and PaIDH2

We assessed the biochemical characteristics of PaIDH1 and PaIDH2 in vitro using Tris-HCl as a buffering agent. The optimal pH for PaIDH1 was determined to be 8.0, slightly higher than that of PaIDH2 (pH 7.5) (Figure 5A). The maximum activity of PaIDH1 was observed at approximately 55 °C, which was higher compared to PaIDH2 (50 °C) (Figure 5B). Furthermore, our thermostability studies revealed that following a 20-min incubation period at 50 °C, PaIDH1 exhibited retention of over 60% of its initial activity, while PaIDH2 displayed retention of only 7% (Figure 5C). At a saturating concentration of the substrate, 2 mM of ATP, ADP, AMP, GTP, GDP, GMP, citrate (CIA), glyoxylate (GLA), α-ketoglutarate (αKG) and oxaloacetic acid (OAA), as well as their mixtures, all exhibited slightly inhibitory effects on the activity of PaIDH1 and PaIDH2 (Table 1). The group compounds of αKG-OAA-GLA at 2 mM caused a reduction in PalDH1 activity to 68% of the original level, whereas the compounds of αKG-OAA-GLA-CIA at 2 mM displayed significant inhibitory activity towards PalDH2 (less than 10%). Although αKG-OAA-GLA-CIA significantly inhibited the activity of PaIDH2, reducing it to 10%, the individual OAA (77.45%) and GLA (93.05%) did not exert a significant effect on PaIDH2 activity. This suggests that the inhibition of PaIDH2 activity occurs when most compounds in the TCA cycle simultaneously act upon it, and it may be a negative feedback mechanism. The effects of αKG-OAA-GLA-CIA, OAA and GLA on PaIDH1 activity amounted to 68.53%, 77.45% and 86.97%, respectively, being weaker than those of PaIDH2. The difference in the activity of PaIDH1 and PaIDH2 affected by the compounds implies that PaIDH1 may have additional regulatory mechanisms to maintain its activity and participate in important biological processes.

The optimal pH for *M. tuberculosis* MtIDH1 and MtIDH2 was found to be 7.5 [34], which was similar to that of PaIDHs, suggesting similarity in their environmental distribution. Compared with PaIDH2, PaIDH1 exhibited greater tolerance to a wider range of pH values and retained over 90% of its activity within the pH range of 7.2–8.5, while the lowest activity observed for PaIDH2 in this range was approximately 70%.

### 2.4. Kinetics Analysis

The catalytic activity of both PaIDH1 and PaIDH2 was observed exclusively towards NADP^+^-linked reactions, while no activity was detected towards NAD^+^-linked reactions, indicating a clear preference for NADP^+^ as the coenzyme (Table 2).

The Km values of PaIDH1 for NADP^+^ and ICT were determined to be 20.00 μM and 14.41 μM, respectively. In comparison to EcIDH (17 μM), both PaIDH1 and AbIDH1 displayed higher Km values for NADP^+^. Interestingly, in terms of the substrate ICT, PaIDH1 demonstrated lower Km values (14.41 μM) compared to EcIDH (24.2 μM) and AbIDH1 (50.5 μM).

The kcat/Km values of PaIDH2 for NADP^+^ and ICT were 4.42 μM^−1^s^−1^ and 7.26 μM^−1^s^−1^, respectively. PaIDH2, MtIDH2 and AbIDH2 were classified as type III-IDH enzymes. Notably, the kcat/Km value of PaIDH2 (4.42 μM^−1^s^−1^) was higher than that of MtIDH2 (1.908 μM^−1^1s^−1^) and AbIDH2 (0.39 μM^−1^s^−1^), which was consistent with the trend observed for substrate ICT as well. Specifically, the kcat/Km value for substrate ICT of PaIDH2 (7.26 μM^−1^s^−1^) was also higher than those of MtIDH2 (1.857 μM^−1^s^−1^) and AbIDH2 (1.9 μM^−1^s^−1^).

### 2.5. Evaluation of Coenzyme Specificity Determinants

Amino acid sequence alignment and crystal structure analysis provided critical insights into the interactions between PaIDH1 and its coenzyme, NADP^+^. Specifically, in PaIDH1, residues Lys346, Tyr347 and Tyr393 engage in direct hydrogen bond interactions with the 3′-phosphate moiety of coenzyme NADP^+^. Comparative analysis between the wild-type PaIDH1 and mutants carrying the K346D and Y393K substitutions demonstrated a remarkable reduction in coenzyme NADP^+^ affinity, by 86-fold and 4-fold, respectively. This observation underscores the pronounced influence of the K346D mutation on coenzyme specificity.

To further explore the impact of these mutations, we introduced a double mutation (K346D/Y347I), which disrupted the interaction between coenzyme NADP^+^ and the protein, resulting in the complete loss of enzymatic activity. This outcome can be attributed to the steric clash between Ile347 and Val353. Both Ile and Val are non-polar hydrophobic amino acids that collectively form a robust hydrophobic core within the substrate-binding pocket of the mutant enzyme. This core effectively hinders coenzyme access to the catalytic center. In response, we strategically substituted Val353 with Ala to mitigate the hindrance imposed by hydrophobic interactions on coenzyme binding.

Subsequent enzyme kinetic analysis of the triple mutation (K346D/Y347I/Val353) revealed an approximate four-fold increase in coenzyme NAD^+^ affinity compared to the single-point mutation (K346D), accompanied by an astounding 50,000-fold enhancement in catalytic efficiency. The four-point mutant PaIDH1-D346I347A353K393 demonstrated even higher affinity for NAD+, with a Km value of 925.1 μM, and totally lost its activity with NADP^+^. These findings suggest that altering the four key amino acid residues resulted in changes in coenzyme dependency (Table 3). As compared with the affinity of PaIDH1 (Km = 20.00 ± 6.29 μM) to NADP^+^, a single-point mutant PalDH1-K393 had lower affinity to NADP^+^ (≈27%), with a Km value of 74.36 ± 5.28 μM, whereas the single-point mutant PaIDH1-D346 exhibited almost no affinity to NADP^+^ (<1.2%), with a Km value of 1715.67 ± 133.5 μM. This result indicates that these two mutations (K393 and D346) do not contribute to NADP^+^ binding, especially Asp at the position 346. The negative charge of D346 probably repels the 2′-phosphate group of NADP^+^ by electrostatic repulsion, resulting in significantly decreased binding affinity toward NADP^+^ and a concomitant increase in affinity for NAD^+^. However, K393 has a positive charge and only decreased the coenzyme NADP^+^-affinity by four-fold. This is in contrast to other natural type I NAD^+^-IDHs, such as ZmIDH (with a Km value of 245 μM) and *Streptococcus suis* IDH (SsIDH) (with a Km value of 233 μM), both of which exhibit higher affinity for NAD^+^ [10,38].

The alignment of the amino acid sequences and analysis of the crystal structure revealed that in PaIDH2, direct hydrogen bond interactions occur between His589, Arg600, Arg649 and the 3′-phosphate moiety of coenzyme NADP^+^. Through site-directed mutagenesis, we generated the single-point mutant enzyme PaIDH2-L589, the two-point mutant enzymes PaIDH2-L589D600 and PaIDH2-L589I600 and the three-point mutant enzymes PaIDH2-L589D600S649, PaIDH2-L589I600D649 and PaIDH2-L589I600L649. Compared to the wild-type PaIDH2, the mutants PAIDH2-L589 and PAIDH2-L589D600 exhibited a 4.5-fold and 28.3-fold decrease in affinity for NADP^+^, respectively, while their catalytic efficiency also decreased by 11.3-fold and 2210-fold, respectively. Additionally, both enzymes showed increased affinity for NAD^+^. However, they still remained dependent on NADP^+^. The mutant PaIDH2-L589D600S649 lost both NAD^+^ and NADP^+^ activity. The Km value of the mutant PaIDH2-L589I600 for NADP^+^ was measured at 7770.67 μM, which was higher than that of NAD^+^ (5824.33 μM) (Table 4), indicating a change in coenzyme specificity to some extent through the modification of these two residues in PaIDH2. Comparing the mutations with the natural type III NAD^+^-IDH, the IDH of Campylobacter exhibited higher affinity for NAD^+^, with a Km value of 88μM [39]. The enzyme kinetics of the PaIDH2-L589I600D649 and PaIDH2-L589I600L649 mutants revealed a loss in NADP^+^ activity, accompanied by significantly reduced affinity for NAD^+^. Moreover, their utilization of NAD^+^ as a coenzyme was found to be minimal (Table 4).

The form of aggregation remained unaltered by the mutation (Appendix A). However, a shift in the coenzyme dependence of PaIDH1 and PaIDH2 was observed following the mutation, resulting in reduced affinity and catalytic efficiency of the mutant enzymes.

### 2.6. In Vitro Phosphorylation and Identification of Phosphorylation Site

In the in vitro phosphorylation experiments, we observed that PaIDH1 was susceptible to phosphorylation, whereas PaIDH2 was not. Phos-tag gel electrophoresis revealed that the lane containing phosphorylated PaIDH1 and PaIDH K/P mixed samples exhibited three distinct bands, while the lane with non-phosphorylated PaIDH2 and PaIDH K/P mixed samples showed only two bands (Figure 6A). Following in vitro phosphorylation reactions, the residual activity of PaIDH1 and PaIDH2 was visualized, as shown in Figure 6B. While the activity of PaIDH2 remained mostly unchanged, that of PaIDH1 was reduced to less than 20% of its original level. Mass spectrometry analysis revealed that Ser115 was the site of phosphorylation on PaIDH1 (Figure 6C).

The amino acid sequence identity between PaIDH1 and EcIDH was approximately 79%. In *E. coli*, Ser113 was identified as the site of phosphorylation for EcIDH [40]. By aligning the amino acid sequences of both proteins, we observed that the phosphorylation site for PaIDH1 (Ser115) and EcIDH (Ser113) was identical and there were conserved residues (Figure 1C), suggesting potential similarity in the phosphorylation mechanisms between *P. aeruginosa* and *E. coli*. In comparison to PaIDH1, it was noteworthy that PaIDH2 lacked the P-loop and AceK recognition segment (ARS), which were recognized by PaIDH K/P, potentially explaining its inability to undergo phosphorylation [35].

## 3. Discussion

The dimeric form of IDH, composed of two identical subunits with a molecular weight of approximately 40–45 kDa, is present in various bacterial species, including *E. coli* [41]. Conversely, the monomeric IDH with a molecular weight ranging from 80 to 100 kDa is exclusively found in *C. glutamicum*, *A. vinelandii* and *Streptomyces avermitilis* [42,43,44]. Only a limited number of species possess both isoforms of IDH, such as *X. campestris*, *M. tuberculosis* and *A. baumannii*; furthermore, the aggregation differs between these two IDH isoforms [34,37,45]. Interestingly, PaIDH1 and PaIDH2 also exhibit distinct aggregation patterns, suggesting the specificity of IDHs in pathogenic bacteria.

Covalent regulation of IDH activity through reversible phosphorylation by IDH K/P is a common mechanism in prokaryotes. Currently, IDH phosphorylation has been extensively investigated in *E. coli*. In this organism, NADP^+^-dependent IDH, IDH K/P and ICL co-regulate carbon flux distribution at branch points. When grown in acetic acid, competition between IDH and ICL for binding to isocitrate resulted in a significant reduction in IDH activity, leading to the accumulation of isocitrate and the initiation of the glyoxylate bypass pathway, which provided precursors for bacterial biosynthesis [46,47]. As previously stated, phosphorylation could not exert regulatory control over carbon flux in the TCA cycle of *M. tuberculosis*, and IDH K/P was absent from this organism [48]. In *M. tuberculosis*, MtIDH1 was dispensable in regulating carbon flow in the TCA cycle and it exhibited greater antigenicity than MtIDH2 [48,49]. In *A. baumannii*, AbIDH1 exerted regulatory control over carbon flux through phosphorylation, whereas AbIDH2 remained unphosphorylated [37]. Due to the phosphorylation of PaIDH1, the enzymatic activity of PaIDH1 was diminished, thereby reducing carbon flux consumption and facilitating the adaptive survival of *P. aeruginosa* on nutrient-limited carbon sources. The regulation mode of PaIDH2 differed from that of PaIDH1, which indicated their involvement in distinct metabolic pathways and thus provided *P. aeruginosa* with greater adaptability to diverse environments for survival and infection.

Our study demonstrated that PaIDHs derived from *P. aeruginosa* PAO1 exhibited thermophilic properties, as they displayed stability at approximately 45 °C, a temperature exceeding ambient conditions. While there were notable similarities in the thermal stability of these two enzymes, certain distinctions were also observed. Notably, at 50 °C, the activity of PaIDH2 was found to be less than 10%, whereas PaIDH1 retained approximately 60% of its activity. The activity of PaIDHs could be effectively inhibited by certain small-molecule compounds, including glyoxylic acid and oxaloacetate. Specifically, these compounds exhibit inhibitory effects on numerous NADP^+^-dependent isocitrate dehydrogenases while sparing NAD^+^-dependent isocitrate dehydrogenases [24]. The distinct enzymatic properties of PaIDH1 and PaIDH2 may lead to their differential adaptation in diverse environments. It is hypothesized that these two enzymes could be differentially expressed across diverse environmental conditions and growth stages.

It has been reported that the coenzyme dependence of two IDH bacteria is mostly conserved, with *M. tuberculosis*, *A. baumannii* [37] and *Colwellia maris* all exhibiting NADP-IDH activity [34,37,50]. Exceptionally, *Xanthomonas campestris* displayed distinct coenzyme dependencies for its two IDH isoforms, where XcIDH1 utilized NAD^+^ as a cofactor and XcIDH2 relied on NADP^+^ [45]. Bacteria harboring NADP-IDH were capable of utilizing acetate as a carbon source, whereas those with NAD-IDH could not support growth due to the insufficient production of NADPH [7,51]. The native EcIDH was NADP-IDH, which was successfully converted into NAD-IDH by substituting crucial amino acid residues [8]. The presence of at least one NADP-IDH isoenzyme in microorganisms with glyoxylate bypass ensured the provision of a major portion of NADPH, which supported bacterial growth on limited carbon sources like acetic acid [7]. However, the decrease in catalytic efficiency observed after changing coenzyme dependence may be attributed to the regulatory role played by the bacteria themselves. This mechanism ensures the sufficient production of NADPH through NADP-IDH activity to sustain survival.

## 4. Materials and Methods

### 4.1. Strains and Reagents

*P. aeruginosa* PAO1 was purchased from the BNCC strain library of the BeNa Culture Collection (BNCC, Beijing). The plasmid *pET-28b* (*+*), *E. coli* DH5α and *E. coli* Rosetta (DE3) were stored in our laboratory. PrimeStar^TM^ HS DNA polymerase was purchased from TaKaRa (Beijing, China). Restriction enzymes and protein molecular weight standards were purchased from Fermentas (Shanghai, China).

### 4.2. Plasmid Construction

Genomic DNA of *P. aeruginosa* PAO1 was extracted by using a bacterial genome extraction kit (TIANGEN, China). The *PaIDH1* (Gene ID: PA 2623) and *PaIDH2* (Gene ID: PA 2624) genes were amplified with the primers in Appendix A. The PCR products with NdeI and XhoI (Thermo Scientific, Shanghai, China) digestion were cloned into expression vector *pET-28b* (*+*) to generate the recombinant plasmids *pET-28b* (*+*)*-PaIDH1* and *pET-28b* (*+*)*-PaIDH2*. Mutations were introduced into PaIDH1 and PaIDH2 by overlap extension, PCR-based, site-directed mutagenesis.

The full-length gene of *IDH K/P* (*AceK*) from *P. aeruginosa* PAO1 was codon-optimized by selecting only the most preferential codons according to the *E. coli* bias and was cloned into the *pET-28b* (*+*) vector with the 6xHis label. GenScript Biotech Corp. (Nanjing, China) performed this procedure.

The correct sequences of each plasmid were verified by a sequencing service (General Biosystems, Hefei, China).

### 4.3. Protein Expression and Purification

*E. coli* Rosetta (DE3) cells harboring *pET-PaIDH1*, *pET-PaIDH2* and *pET-PaIDH K/P* were cultured overnight with 30 ug/mL kanamycin and 30 ug/mL chloramphenicol and then incubated in 200 mL fresh LB medium with the same antibiotic at 37 °C. When the OD_600_ of the cultures reached 0.4–0.6, isopropyl-1-thio-β-d-galactopyranoside was added to the cultures at a final concentration of 0.5 mM with subsequent cultivation 20 h at 20 °C. Cells were harvested by centrifugation at 5000 rpm for 5 min and then resuspended in lysis buffer (50 mM sodium dihydrogen phosphate, 300 mM NaCl, pH 7.5). The insoluble debris was removed by centrifugating at 10,000 rpm for 20 min at 4 °C. Then, enzymes with the 6×His-tag were purified using BD TALON Metal Affinity Resins (Dalian, China) according to the manufacturer’s instructions. The expression abundance and purification homogeneity were verified by sodium dodecyl sulfate (SDS)–polyacrylamide gel electrophoresis (PAGE). Mutants PaIDH1 and PaIDH2 were expressed and purified as the wild-type PaIDH1 and PaIDH2.

### 4.4. Enzyme Assay and Kinetic Studies

The activity of PaIDH1 and PaIDH2 was routinely assayed at 25 °C and pH 7.5 by monitoring the increase in NADPH at 340 nm with a thermostat, the Cary 300 UV–Vis spectrophotometer (Varian, CA, USA), using a molar extinction coefficient of 6.22 mM^−1^cm^−1^. Standard reaction mixtures contained 50 mM Tris-HCl buffer (pH 7.5), 2 mM MnCl_2_, 1 mM DL-isocitrate and 500 μM NADP^+^ (NAD^+^). The minimum amount of enzyme required to stabilize the reaction was utilized. One unit of enzyme activity represented the formation of 1 μM of NADPH(NADH) per minute. Protein concentrations were determined using the Bio-Rad protein assay kit (Bio-Rad, CA, USA) with bovine serum albumin as the standard. To measure the Michaelis constant (*K*_m_) values of PaIDH1 and PaIDH2 for NADP^+^(NAD^+^), the isocitrate concentration was kept fixed at 1.0 mM with varying cofactor concentrations. To measure the Michaelis constant (*K*_m_) values of PaIDH1 and PaIDH2 for ICT, the NAD(P)^+^ concentration was kept fixed at 500 μM with varying substrate concentrations. Apparent maximum velocity (V_max_) and *K*_m_ values were calculated by nonlinear regression using Prism 7.0 (Prism, CA, USA).

### 4.5. Effects of pH, Temperature and Compounds

In the experiments in which the optimal pH was tested, the activity of purified recombinant PaIDH1 and PaIDH2 was determined in a 50 mM Tris-HCl buffer between pH 6.8 and 8.8 using the reaction mixtures described above. For temperature profile analysis, enzymes were assayed at various temperatures from 25 to 60 °C. To estimate thermal stability, enzymes were incubated at 25–55 °C for 20 min; after incubation, the enzymes were immediately put on ice for 5 min, and the residual enzymes’ activity was measured using the above standard enzyme assay reaction mixtures. We added ATP, ADP, AMP, GTP, GDP, GMP, citrate, glyoxylate, α-ketoglutarate, oxaloacetate and their mixtures to the enzyme reaction system, mixed them with the system, added the enzyme, and then the effect of the small molecules on enzyme activity was determined by detecting the production of products. The compounds were added at a consistent final concentration of 2 mM, while the detection process was carried out under conditions ensuring the complete saturation of both substrate and coenzyme.

### 4.6. Gel Filtration Chromatography

The molecular masses of PaIDH1 and PaIDH2 were estimated by gel filtration chromatography on a Hiload^TM^ 10/300 Superdex 200 column (GE Healthcare Life Sciences, Pittsburgh, PA, USA), equilibrated with 0.05 M potassium phosphate buffer (pH 7.0) containing 0.15 M NaCl. Protein standards for the calibration of gels were Ovalbumin (45 kDa), Conalbumin (75 kDa), Aldolase (158 kDa), Ferritin (440 kDa) and Thyroglobulin (669 kDa). The formula for calculation was lgMW = 7.7986 − 0.206 Ve. The protein sample volume was approximately 0.5 mL, containing approximately 4 mg of protein.

### 4.7. In Vitro Phosphorylation

The phosphorylation reaction system (500 μL) contained 25 mM Tris-HCL (37 °C, pH 7.5, 100 mM NaCl), 5 mM MgCl_2_, 200 μM ATP and enzymes. The concentration of PaIDH K/P and IDH used in running the gel was 1:1. Then, the operation process was similar to that of ordinary SDS-PAGE, except that metal ions (Mn^2+^) and Phos-tag^TM^ (Boppard, Hongkong, China) acrylamide were added during the preparation of the SDS-PAGE separation gel. Phos-tag^TM^ is a functional molecule that specifically binds to phosphate ions. It can be used for the separation of phosphorylated proteins. During protein electrophoresis, phosphorylation groups specifically bind to Phos-tag^TM^ acrylamide chelated with metal ions, resulting in the slower migration of phosphorylated proteins. After running gel, the phosphorylated bands were cut from the gel for mass spectrometry analysis (APT, Shanghai, China). For the determination of the residual activity of the enzyme after phosphorylation, we utilized the phosphorylation reaction system and measured the enzyme activity immediately after the end of the 37 °C water bath. The concentration of PaIDH K/P and IDH used in the determination of residual activity was 1:2.

## 5. Conclusions

In this study, two mutants, PaIDH1-D346I347A353K393 and PaIDH2-L589I600, were successfully engineered to exhibit an alteration in their coenzyme specificity from NADP^+^ to NAD^+^, elucidating the coenzyme evolution pathway of *P. aeruginosa* IDHs. Meanwhile, we found that PaIDH1 and PaIDH2 had different regulatory patterns and enzymatic properties, implying that they may participate in different metabolic pathways, thus making *P. aeruginosa* more adaptable to survival and infection in different environments.

## Figures and Tables

**Figure 1 ijms-24-14985-f001:**
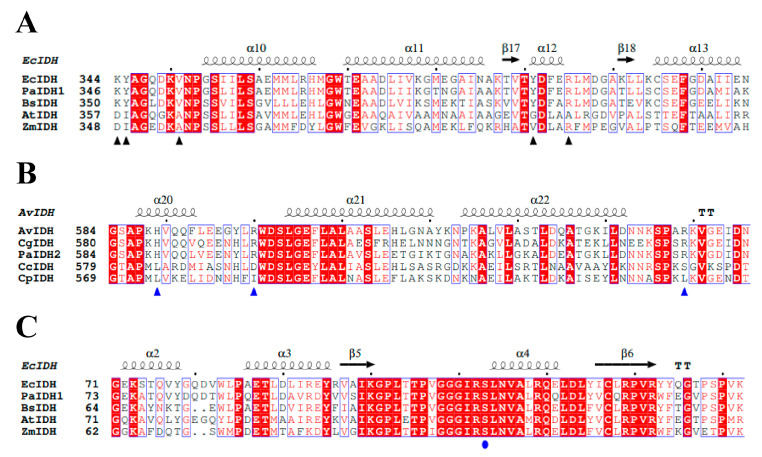
Structure-based protein sequence alignment of PaIDH with other IDHs. Putative coenzyme binding sites are marked with triangles. Phosphorylation site of PaIDH1 is marked with circle. The completely conserved residues are shaded in red. The secondary structure elements are placed above the alignment. Image created by ESPript 3.0 [24]. (**A**) The deduced conserved residues involved in PaIDH1 coenzyme binding were aligned with prokaryotic NADP-IDHs (*Escherichia coli* IDH and *B. subtilis* IDH) and prokaryotic NAD-IDHs (*Acidithiobacillus thiooxidans* IDH and *Zymomonas mobilis* IDH) based on the PaIDH1 structure (PDB entry 5m2e) and AtIDH structure (PDB entry 2d4v). (**B**) The deduced conserved residues involved in PaIDH2 coenzyme binding were aligned with prokaryotic NADP-IDHs (*Azotobacter vinelandii* IDH, *Corynebacterium glutamicum* IDH and *Acinetobacter baumannii* IDH2) and prokaryotic NAD-IDHs (*Campylobacter curvus* IDH and *Campylobacter pinnipediorum* IDH) based on the PaIDH2 structure (PDB entry 6g3u). (**C**) Comparison of PaIDH1 phosphorylation sites with other prokaryotes based on the PaIDH1 structure (PDB entry 5m2e).

**Figure 2 ijms-24-14985-f002:**
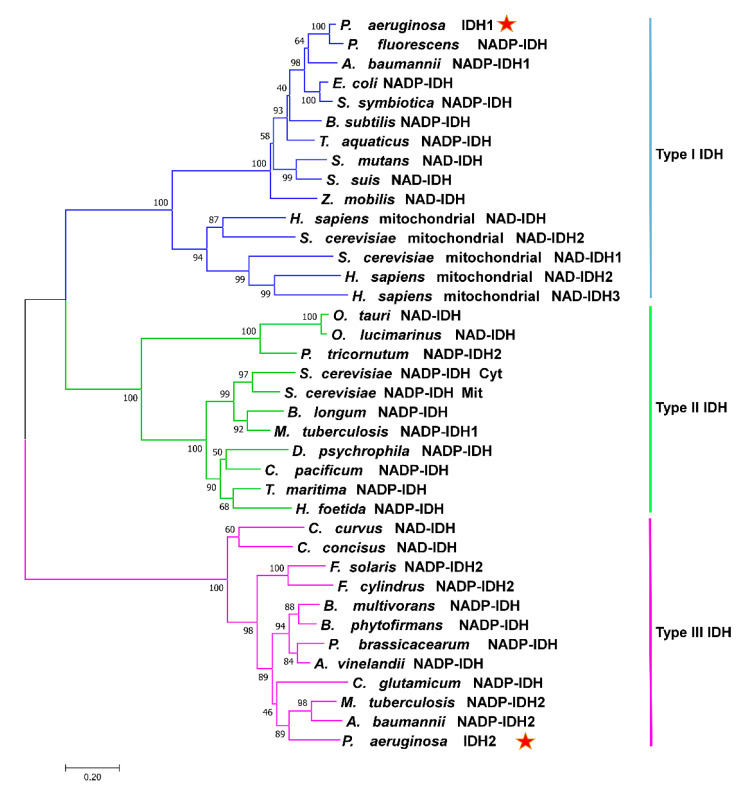
Phylogenetic analysis of IDHs from different species. The analysis involved 38 IDH sequences and a neighbor-joining tree with 1000 bootstraps and was created by MEGA 7.0. PaIDHs are marked by red stars.

**Figure 3 ijms-24-14985-f003:**
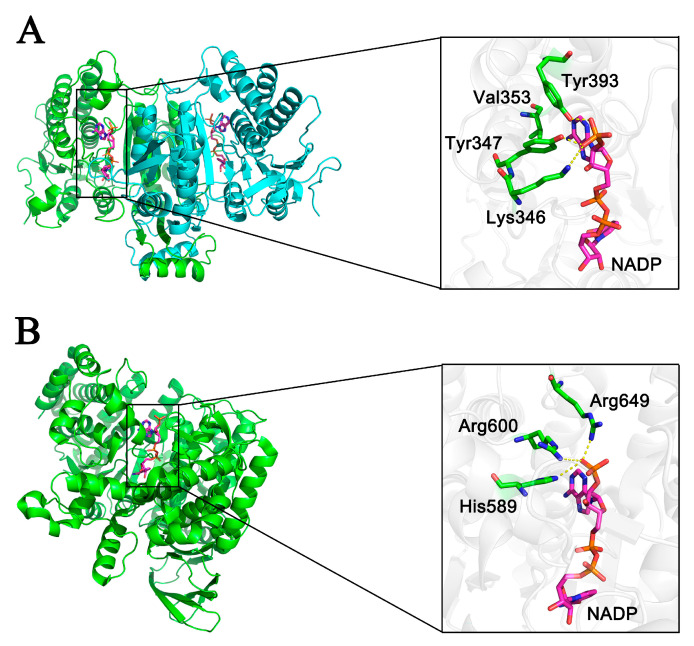
Molecular modeling of PaIDH1 and PaIDH2. (**A**) Binding model of PaIDH1 with coenzyme NADP^+^. On the left, a superimposition illustrates the overall structure of modeled PaIDH1 in green, with the PaIDH1 model generated using the Swiss-model server and *E. coli* IDH in cyan (PDB code: 4aj3) as a template. The magnified view on the right highlights critical determinants of coenzyme specificity, depicted as thick sticks. Site-directed mutagenesis targeted Lys346, Tyr347, Val353 and Tyr393. (**B**) Binding model of PaIDH2 with coenzyme NADP^+^. On the left, a superimposition presents the overall structures of modeled PaIDH2 in green (PDB code: 6g3u). The enlarged view on the right highlights essential determinants of coenzyme specificity, represented as thick sticks. Site-directed mutagenesis focused on His589, Arg600 and Arg649.

**Figure 4 ijms-24-14985-f004:**
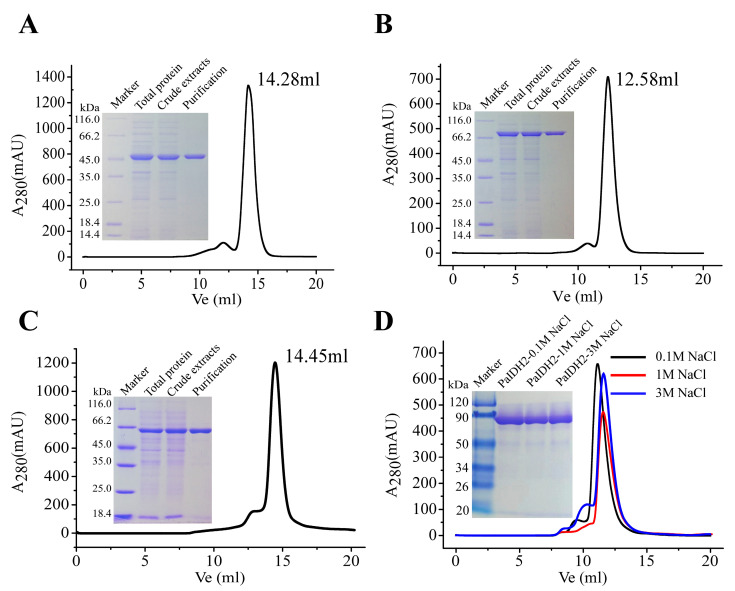
Determination of the molecular mass of PaIDH1 (**A**), PaIDH2 (**B**,**D**) and PaIDH K/P (**C**). The flow rate was 0.5 mL/min and the proteins in the fractions were monitored at 280 nm. (**A**) Result of gel filtration chromatography of PaIDH1. (**B**) Result of gel filtration chromatography of PaIDH2. (**C**) Result of gel filtration chromatography of PaIDH K/P. (**D**) Result of gel filtration chromatography of PaIDH2 at different salt concentrations. The elution volume was observed to be 11.12 mL at a NaCl concentration of 0.1 M, 11.57 mL at a NaCl concentration of 1 M and 11.75 mL at a NaCl concentration of 3 M. The experiment was conducted three times in accordance with the principles of biology.

**Figure 5 ijms-24-14985-f005:**
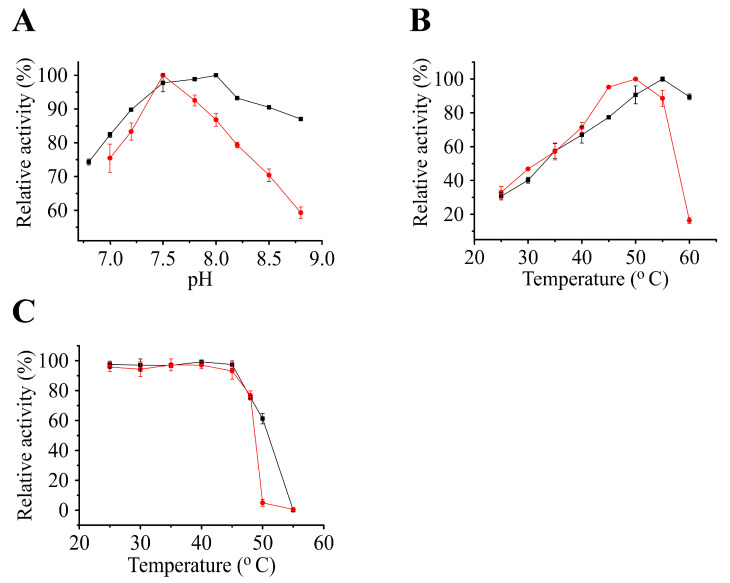
Effects of pH and temperature on the activity of purified PaIDH1 and PaIDH2. (**A**) Effects of pH range of 6.8–8.8 on PaIDH1 (■) and PaIDH2 (●), respectively. (**B**) Effects of temperature range of 25–60 °C on PaIDH1 (■) and PaIDH2 (●), respectively. (**C**) Heat inactivation profiles of PaIDH1 and PaIDH2. The PaIDH1 (■) and PaIDH2 (●) activity was measured from 25 to 55 °C, respectively.

**Figure 6 ijms-24-14985-f006:**
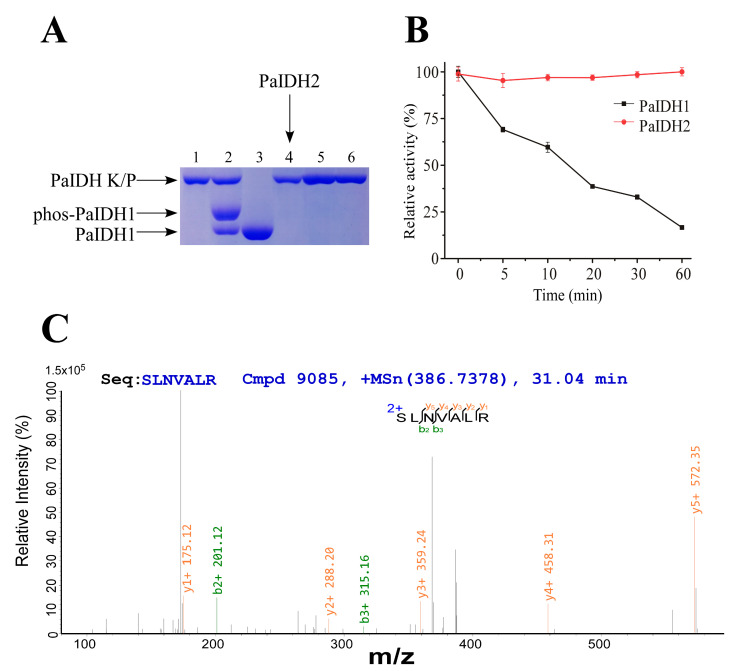
Identification of the phosphorylated PaIDH1 and PaIDH2. (**A**) Identification of the phosphorylated PaIDH1 and PaIDH2 in vitro using Phos-tag SDS-PAGE. Lane 1, 6: PaIDH K/P as the control. Lane 2: PaIDH1 treated with PaIDH K/P. Lane 3: PaIDH1 untreated with PaIDH K/P as the control. Lane 4: PaIDH2 untreated with PaIDH K/P as the control. Lane 5: PaIDH2 treated with PaIDH K/P. (**B**) In vitro phosphorylation of PaIDH1 and PaIDH2 by EcIDH K/P and PaIDH K/P. (**C**) Mass spectrometry detection of PaIDH1 phosphorylation site. (S) indicates a phosphorylation site.

**Table 1 ijms-24-14985-t001:** Effects of different metabolites on the activity of PaIDH1 and PaIDH2.

Metabolite	Relative Activity (%)	
	PaIDH1	PaIDH2
Control (none)	100 ± 0.00	100 ± 0.00
ATP	89.16 ± 5.17	75.91 ± 5.92
ADP	81.79 ± 3.55	82.66 ± 3.37
AMP	87.30 ± 1.95	91.17 ± 6.95
ATP + ADP	79.04 ± 7.50	52.48 ± 4.76
AMP + ADP + AMP	68.07 ± 6.59	59.61 ± 5.56
ATP + AMP	83.89 ± 3.81	51.02 ± 5.04
ADP + AMP	81.15 ± 5.73	80.68 ± 7.31
GTP	85.13 ± 1.48	93.99 ± 1.3
GDP	86.69 ± 1.16	88.43 ± 2.80
GMP	83.32 ± 2.58	93.26 ± 3.09
GTP + GDP	85.97 ± 1.38	90.88 ± 2.47
GTP + GMP	85.74 ± 2.14	89.83 ± 4.12
GDP + GMP	86.79 ± 1.66	87.63 ± 3.59
GTP + GDP + GMP	84.43 ± 1.59	79.25 ± 4.96
CIA	86.97 ± 4.95	94.81 ± 4.82
GLA	93.05 ± 3.88	87.06 ± 3.73
α-KG	79.52 ± 4.64	82.79 ± 5.65
OAA	77.45 ± 4.56	88.32 ± 7.97
α-KG + OAA	71.54 ± 4.76	70.7 ± 2.97
GLA + OAA	75.71 ± 6.02	79.88 ± 7.08
α-KG + OAA + GLA	68.17 ± 4.42	16.23 ± 0.56
α-KG + OAA + GLA + CIA	68.53 ± 0.86	9.926 ± 1.22

Data are the mean ± SD of at least three independent measurements.

**Table 2 ijms-24-14985-t002:** Comparison of prokaryotic Michaelis–Menten constants.

Enzymes	ICT			NADP^+^			References
	*K*_m_ (μM)	*k*_cat_ (s^−1^)	*k*_cat_/*K*_m_ (μM^−1^s^−1^)	*K*_m_ (μM)	*k*_cat_ (s^−1^)	*k*_cat_/*K*_m_ (μM^−1^s^−1^)	
PaIDH1	14.41 ± 1.82	29.26 ± 10.92	2.10 ± 0.93	20.00 ± 6.29	104.51 ± 19.91	5.42 ± 0.97	This study
PaIDH2	23.61 ± 1.21	171.34 ± 6.77	7.26 ± 0.13	30.68 ± 2.73	133.98 ± 17.04	4.42 ± 1.13	This study
EcIDH	24.2 ± 6.6	51.8 ± 4.4	2.1	17	80.5	4.7	[8]
MtIDH1	10 ± 5	3.8	0.38	125 ± 5	4	0.032	[34]
MtIDH2	20 ± 1	37.13	1.857	19.6 ± 6	37.4	1.908	[34]
AbIDH1	50.5 ± 3.2	50.4	1.0	46.6 ± 6.1	60.1 ± 3.3	1.3	[37]
AbIDH2	21 ± 3	39.2 ± 2.1	1.9	94 ± 6	36.9 ± 1.2	0.39	[37]

Data are the mean ± SD of at least three independent measurements.

**Table 3 ijms-24-14985-t003:** Kinetic determination of the wild-type and mutant PaIDH1.

Enzymes	NADP^+^			NAD^+^		
	*K*_m_ (μM)	*k*_cat_ (s^−1^)	*k*_cat_/*K*_m_ (A) (μM^−1^s^−1^)	*K*_m_ (μM)	*k*_cat_ (s^−1^)	*k*_cat_/*K*_m_ (B) (μM^−1^s^−1^)
PaIDH1 PaIDH1-D346 PaIDH1-D346I347 PaIDH1-D346I347A353 PaIDH1-D346I347A353K393 PaIDH1-K393	20.00 ± 6.29 1715.67 ± 133.5 - - - 74.36 ± 5.28	104.51 ± 19.91 39.56 ± 2.43 - - - 41.32 ± 1.95	5.42 ± 0.97 0.02 ± 0.003 - - - 0.56 ± 0.04	- 15,368.79 ± 2818 - 4500 ± 157.36 925.1 ± 83.81 -	- 0.003 ± 0.0003 - 4.73 ± 0.76 7.04 ± 0.71 -	- 0.02 × 10^−6^ - 0.001 ± 0.0002 0.008 ± 0.001 -

-: Not detected. Data are the mean ± SD of at least three independent measurements.

**Table 4 ijms-24-14985-t004:** Kinetic determination of the wild-type and mutant PaIDH2.

Enzymes	NADP^+^			NAD^+^		
	*K*_m_ (μM)	*k*_cat_ (s^−1^)	*k*_cat_/*K*_m_ (A) (μM^−1^s^−1^)	*K*_m_ (μM)	*k*_cat_ (s^−1^)	*k*_cat_/*K*_m_ (B) (μM^−1^s^−1^)
PaIDH2	30.68 ± 2.73	13 3.98 ± 17.04	4.42 ± 1.13	-	-	-
PaIDH2-L589	135.33 ± 14.4	52.36 ± 9.08	0.39 ± 0.1	5927.3 ± 205.25	10.89 ± 0.15	0.002 ± 0.00008
PaIDH2-L589 D600	849.33 ± 71.92	1.7 ± 0.35	0.002 ± 0.0004	8949.33 ± 748.06	4.5 ± 0.41	0.0005 ± 0.000003
PaIDH2-L589 D600 S649	-	-	-	12242.33 ± 497.22	6.63 ± 0.12	0.0005 ± 0.00001
PaIDH2-L589 I600	7770.67 ± 270.55	16.03 ± 0.79	0.002 ± 0.00005	5824.33 ± 346.01	20.74 ± 0.43	0.004 ± 0.0002
PaIDH2-L589 I600 L649	-	-	-	15625.33 ± 1751	28.92 ± 2.73	0.002 ± 0.0001
PaIDH2-L589 I600 D649	-	-	-	9968.67 ± 1051.5	48.19 ± 4.15	0.005 ± 0.0002

-: Not detected. Data are the mean ± SD of at least three independent measurements.

## Data Availability

Data could be found in the Appendix A.

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
