# Peer review of "Two Different Isocitrate Dehydrogenases from Pseudomonas aeruginosa: Enzymology and Coenzyme-Evolutionary Implications"

_ijms, 2023, doi:10.3390/ijms241914985_

Round 1

Reviewer 1 Report

Major points

-       Page 4 line 130 and 135. Authors suggest that PaIDH1 has a molecular mass of 75.4kDa with an elution volume of 14.28. However, unlike the value indicated in figure 3A, the main elution pick seems to be around 12.5ml. Similarly, authors affirm that PaIDH2 is present in solution as a trimer or elongated dimer. However, in the fig 3B the indicated elution volume of 12.58 ml is clearly underestimated. In fact, the main elution pick in the figure 3B has an elution volume around 14 ml. Authors should explain these discrepancies.

-          The rational of SEC analysis of the PaIDH2 at different ionic strength should be better explained and some data have to be indicated, e.g. (a) the elution volumes at different salt concentrations, (b) the concentration of the injected protein.

-          Authors suggest the presence of a tetrameric PaIDH2 at 0.3M (NaCl?) concentration that is not present at higher salt concentration (Page 4, line 150). Authors should clearly indicate in the picture the tetramer elution pick. Moreover, legend in fig3D indicates 0.1M NaCl, while in the text is reported 0.3M, which is the salt concentration used?

-          The effect of compounds on the catalytic activity of PaIDH1 and 2, is not properly described. The concentration(s) of each tested compound should be indicated as well as the assay conditions. (i) Was each compound tested at single concentration? (ii) If yes, how each concentration was selected? (iii) were the assays performed in the presence of saturating or under-saturating concentration of the substrate?

Moreover, the strong effect of the combination of αKG-OAA-GLA-CIA on PaIDH2 activity with respect OAA and GLA alone, and their differential effect on PaIDH1 and 2,  should be discussed.

-          Page 8, line 232. Asp 346, Ile347, Ala353 and Lys 393 are the residues inserted by site directed mutagenesis, not the original ones of the putative coenzyme binding domain of PaIDH1. It would be more appropriate to present the original coenzyme binding residues of the PaIDH1 and then explain the rational of the selected substitutions.

     As a suggestion, the available of the crystal structures can be used to generate very illustrative 3D images of the putative coenzyme binding domain of PaIDH1 and 2, that could be helpful for the reader.

-          Authors should discuss also the effects of the PaIDH1-K393 mutation

-          Considering that the mutations A353-K393 together consistently increase the PaIDH1 affinity for the NAD+ coenzyme, why the authors didn't analyze the mutant PaIDH1-A353K393 

-          Page 10 line 308, authors affirm that Km value of PaDH1 (20.0 ± 6.3 µM) is lower than that of PaIDH2 (30.7 ± 2.7 µM). However, considering the limits of errors, the two values are not significantly different.

-          Page 12 lines 272-374. In my opinion, in the absence of in-depth structural studies or in silico predictions about the effects of the mutations, the suggested relation between the catalytic efficiency decrease of the variants and an inherent protective mechanism of the bacteria seems to be too speculative.

Minor points

-          Page 1, line 26. As a suggestion, in this case “oligomeric state” would be more appropriate than “polymerization state”

-          Details and text in Fig 5C are not clearly readable. The image should be improved

-          All the complete names of the acronyms reported in figure 1 should be specified in the legend (e.g. AbIDH1, EcIDH)

-          Compound abbreviations such as ICL, GLA, CIA etc…should be indicated with the full name when reported for the first time

-          In the materials and methods section, the following information should be added:

(i)                  The expression vector used to express the IDH K/P (Was IDH K/P equipped with a 6xHis-tag?)

(ii)                 The method/service used to verify the DNA sequence of the mutated expression vectors

(iii)               The enzyme assays volume

(iv)               Details about the assays in presence of different compounds (see on the major point)

(v)                The equation of the calibration curve used to calculate the molecular weight by SEC analysis

(vi)               The volume of injection, the protein concentration and the flow rate used on SEC

Reviewer 2 Report

The authors of the manuscript have deliberated on the biochemical characterization of the isoenzymes of isocitrate dehydrogenase from Pseudomonas aeruginosa and they have provided some insights into the evolutionary origins of coenzyme specificity of the enzymes. The experiments are designed well, and the conclusions are supported by the data. The manuscript can be accepted for publication, however, there are a few issues that the authors should address.

1.     Lines 80-82: ‘In this study, we cloned and overexpressed genes encoding IDH1, IDH2, and IDH K/P from P. aeruginosa PAO1 as fused proteins in E. coli’. The authors must elaborate what they mean by ‘fused proteins’.

2.     Lines 152-154: The authors have alluded to ‘distinct polymerization forms of PaIDH2 in response to changes to ionic strength and the relationship between these distinct protein conformations remains elusive’. Have the authors attempted to assay the activities of PaIDH2 in the presence of different salt concentrations? Does the co-factor specificity change with the changes in conformation?

3.     In Figure 4D, the authors do not mention the change in molecular mass of IDH2 at each of the salt concentrations. Also, the shift in retention volume of PaIDH2 from 0.1 - 1M NaCl, appears to be quite marginal. Is this result reproducible? The authors should mention the number of replicate SEC measurements performed at the different ionic strengths.

4.     Section 2.5: The first sentence of this section (lines 231-232) reads ‘According to the structure-based sequence alignment (Fig. 1A), it was hypothesized that Asp346, Ile347, Ala353, and Lys393 were putative coenzyme binding sites of PaIDH1.’ The sentence appears to highlight that Asp346, Ile 347, Ala 353, and Lys393 are the residues in the wildtype enzyme that bind to the co-factor. However, from the primer sequences in the supplementary information, the residues in the coenzyme binding sites of wildtype PaIDH1 are K346, Y347, V353, Y393 and these are substituted with Asp, Ile, Ala and Lys, respectively. The authors should reword the sentences in lines 231-232 to mention the residues of wildtype enzyme that are substituted. The sentences 251-252 in section 2.5 should also be reworded in a similar manner.

5.     In tables 3 and 4, the authors have reported that no activity could be detected for a few of the IDH mutants. Did the authors attempt to assay the activities at higher concentrations of these mutants? If the activities are measurable upon increasing the concentration of the mutant enzymes, the authors should report the corresponding Km, kcat and kcat/Km values.

1.     Following are a few typographical errors in the manuscript.

a.     Line 127: Correct ‘Supernuant’ to ‘Supernatant’

b.     Line 402: Correct ‘isopropy’ to isopropyl 

c.     Line 444: Correct ‘Tirs’ to ‘Tris’

Round 2

Reviewer 1 Report

Minor points

- The elution profile in Fig4A and 4B are still inverted. In fact, on the basis of the X-axis volume scale, the elution pick in Fig4A is around 12.5 ml instead of 14.28 ml as indicated on the top of the pannel A (and vice versa for the Fig4B)

- Figure 3 legend. (i) Panel A, please indicate in the legend the name of the structure represented in cyan color. 

- Page 14, line 450. "The correct recombinant plasmids and mutates, verified by sequencing" should be "The correct sequence of each plasmid was verified by sequencing service (...)" 
